# Integrating geospatial tools in mapping forest fire severity and burned areas in the Western Usambara Mountain Forests, Lushoto, Tanzania

Braison P. Mkiwa [ID]1*, Ernest W. Mauya[2], Justo N. Jonas [ID]1, Gimbage E. Mbeyale [ID]3

1 Department of Ecosystems and Conservation, College of Forestry, Wildlife and Tourism, Sokoine University of Agriculture, Morogoro, Tanzania, 2 Department of Forest Engineering and Wood Sciences, College of Forestry, Wildlife and Tourism, Sokoine University of Agriculture, Morogoro, Tanzania, 3 Department of Forest Resource Assessment and Management, College of Forestry, Wildlife and Tourism, Sokoine University of Agriculture, Morogoro, Tanzania

* mkiwabraison@yahoo.com

## Abstract

Despite the numerous negative effects of tropical forest fires in Tanzania, the sources and effects remain insufficiently documented. This study aimed to develop an integrated approach that combines geospatial tools and socio-economic data to assess the sources and effects of forest fires and map burn severity and its trends over 10 years in West Usambara Mountain Forests. Three approaches including Participatory Rural Appraisal (PRA), satellite image analysis, and direct observation were used to generate information on spatial and temporal forest fire severity. Findings revealed that farm preparation (38.2%) and charcoal preparation (21.2%) are the primary source of these forest fires. Burn severity maps showed 32.12% to 20.31% of combined high and low severity areas, with a total burned area of 3,296.96 hectares, accounting for 15.86% of the reserves. The differenced Normalized Difference Vegetation Index (dNDVI) maps revealed 36.30% to 21.10 of high and low severity areas, while post-fire NBR and NDVI time series indicated a significant vegetation loss (0.21 to 0.36). This study demonstrates the integration of remote sensing and socio-economic approaches to enhance forest fire management, conservation, policy enforcement, and community awareness that can be upscaled to other forest areas for effective management.

## Introduction

Tropical forests are recognized worldwide for their importance in protecting and preserving various ecosystem services. For example, according to [1], tropical forests alone store a quarter of a trillion tons of carbon above and below biomass [2]. Unfortunately, they are deliberately affected by forest fire dynamics, causing approximately 30% of degradation in tropical forests caused by high temperatures and low humidity

**Data availability statement:** "All relevant data are within the paper and its Supporting Information files. Some raw data are available free of charge to all users on the Google Earth Engine (GEE) platform and can be obtained here: https://code.earthengine.google.com/. Otherwise, raw data can be made available by the corresponding author upon reasonable request."

**Funding:** B.P. Mkiwa - Name of the author received the award Grant number: 2022/RP/M000710160 Name: Tanzania Forest Fund (TaFF) TaFF URL:https://www.mfukowamisitu.go.tz The funder did not play any role in the study design, data collection and analysis, decision to publish, or manuscript preparation.

**Competing interests:** I have read the journal's policy and the authors of this manuscript have the following competing interests: Reports of financial support provided by the Tanzania Forest Services Agency (TFS) and Tanzania Forest Fund (TaFF).

[3]. With increased forest fires in recent years, the flora and fauna in these ecosystems have been adversely affected in multiple ways through human and natural activities [4].

Shifting cultivation remains the main practice of indigenous people in many tropical parts of the world [5]. Likewise, the Convention on Biological Diversity (CBD), Aichi Target 11 emphasizes the necessity for effective and equitable management of protected areas [6], highlighting the importance of understanding local community perceptions regarding the sources and effects of forest fires since perceptions can significantly influence the success of conservation efforts [7]. Burned forests become severely damaged, causing the loss of native natural forests, introduction of invasive species that pose a significant threat to the native species, and ultimately cause massive economic loss to the people [8]. The region is now more vulnerable to forest fires during the dry season than in the past due to climate change, increased human activities, and the presence of elevated areas [9,10].

Forest fires are a major problem and protecting them is our concern [11] as they are linked directly or indirectly to intentional or unintentional human factors [8]. A forest fire, also known as a wildfire or bushfire is an uncontrolled and unplanned combustion of vegetation in a natural setting such as forests or grasslands, while Burn severity is defined as the level of soil and vegetation change after a fire [12]. In Tanzania like many countries in sub-Saharan Africa, experiences an annual burning of about 403,400 ha of land, with approximately 12% of its land lost to fire each year between 2001 and 2007 [13,14], ranking the country fourth in SADC [15]. Miombo woodlands contribute to 75% of these fires, followed by forest plantations (20%), and forest reserves (5%) [11].

On the other hand, due to the climate changes that are currently facing the world, tropical montane forests such as Magamba Nature Forest Reserve (MNFR), Mkusu Forest Reserve (MFR), and Shagayu Forest Reserve (SFR) in the Western part of EAMs of Lushoto District in Tanzania are also facing a similar problem of frequent forest fires. It is reported that between 1997/1998, 2016, and 2021/2022 about 6,110 ha in the total area were burned in Magamba NFR [16] while, about 210 ha, and 120 ha for MFR and SFR respectively also were burned in 2021/2022 [17]. This indicates that forest fires significantly impact forests and can cause severe substantial environmental harm if not properly managed [11].

Despite the numerous negative effects of fire, the recent comprehensive research on the sources, and effects of forest fires in tropical rainforests, particularly in Tanzania's West Usambara of EAMs remains insufficiently documented [14,18]. Furthermore, preventive and mitigative measures for minimizing the ever-increasing threat of forest fires are inadequate to identify Tanzania's long-term fire monitoring programs through mapping, size estimation, and distribution of forest fires [5,15]. In particular, uncontrolled fires in tropical EAMs pose a significant threat because the plants in these ecosystems are not adapted to fire events as a result promote biological invasion that sometimes are extremely flammable and prone to fire, for example, the Blacken ferns (*Pteridium aquilinum*) which immediately become dominant in burn areas after the fire [8,19]. Therefore, mapping forest fires and burned areas is crucial

for understanding their sources and effects, informing management strategies, monitoring trends, engaging communities, and supporting policy development [9]. This integrated approach enhances the ability to manage forest fires effectively while promoting sustainable conservation.

Forest-based activities in Tanzania, particularly in Lushoto District, are predominantly traditional and encompass fuel-wood collection, charcoal production, and tree planting in woodlots [9,14]. These practices have created a mutually benefi-cial scenario for conserving natural resources while also supporting local communities, especially in the tourism sector, and enhancing water availability [9,20]. However, the spatial and temporal patterns of forest fires and their underlying drivers in the forest sector remain unclear due to a significant lack of historical fire records in Tanzania and limited spatial coverage and replication [9,21]. Satellite image data indicates that approximately 11 million ha burn annually in protected areas across Tanzania, particularly in the West, Southern, and Eastern blocks of the Eastern Arc Mountain Forests [9,20].

Remotely sensed data have contributed to the increased speed, cost efficiency, precision, and timeliness associated with inventories and made it possible to generate maps of forest characteristics with different spatial resolutions and accuracies than before [22,23]. Because most forest fires occur in remote areas, they go undetected as a result, remotely sensed satellite data is the best alternate source for forest fire studies [5]. So through RS, continuous information over large forest areas being affected by forest fires in terms of area and time was assessed using near-infrared (NIR) and short-wave infrared (SWIR) bands from pre- and post-fire satellite images [24]. Likewise, the Landsat and Sentinel-2 satellites offer advantages in forest fire detection, identification, and mapping of forests, with this need for monitoring and reporting due to their spatial and temporal resolution [25]. Remote sensed satellite data are used to create indices that indicate many aspects of the earth's surface, such as vegetation, temperature, and humidity [5]. The Normalized Differ-ence Vegetation Index (NDVI) is a potential index for assessing vegetation status, detecting burnt areas, and changing flora due to forest fires [26]. The Nomalize Burn Ratio (NBR) and differenced NBR (dNBR) are also included in this study since are widely used to infer fire severity from remotely sensed data [27].

The study aims, to develop an integrated approach of using geospatial tools and social-economic data to assess forest fire severity and accurately map burned areas in West Usambara Mountain Forests, exploring (i) What is the spatial distribution of forest fire hotspots and areas affected by varying degree of burn severity (ii) How have forest fire trends changed over the past 10 years, and iii) What are stakeholders' perception regarding the sources and effects of forest fires in the region. The study starts by identifying and assessing forest fire severity and burned areas by mapping them through remote sensing technology over the last 10 years from 2013 to 2023, defining all the possible sources, and effects that arise from forest fires. Afterward, the findings will be presented to the forest departments to adopt potential preventive measures and policymakers which is useful in deciding the problem of forest fires.

## Materials and methods

### Study area

The study area covers three locations namely MNFR, MFR, and SFR (Fig 1) which are part of EAM in Usambara Western Block Mountain Forests. MNFR is found in both Lushoto and Korogwe, whereas a smaller part is located in Korogwe [28]. The district was chosen based on the prevalence of fire incidences during dry season and because the local communities are actively involved in protecting the forests against destruction by forest fires [29]. Geographically, the district is situated in the Northern part of Tanga region between 4° 57' 54" latitude South of equator and 38° 30' 51" longitude East of Green-wich [16].

The study areas are located in the Tropical climatic zone and ranges in altitude from 1000 m to 2100 m with a total area of 20,787 ha collectively. The District receives rainfall on a bimodal pattern, with short rains from October to December, and long rains from March to June with annual rainfall ranging from 800–2000 mm per year [28]. Temperature ranges from 15 $^0$C to 30 $^0$C annually [16]. The soils are classified as Luvic phaezem, Chromic Luvisol, Mollic Glaysol, and Rhodic

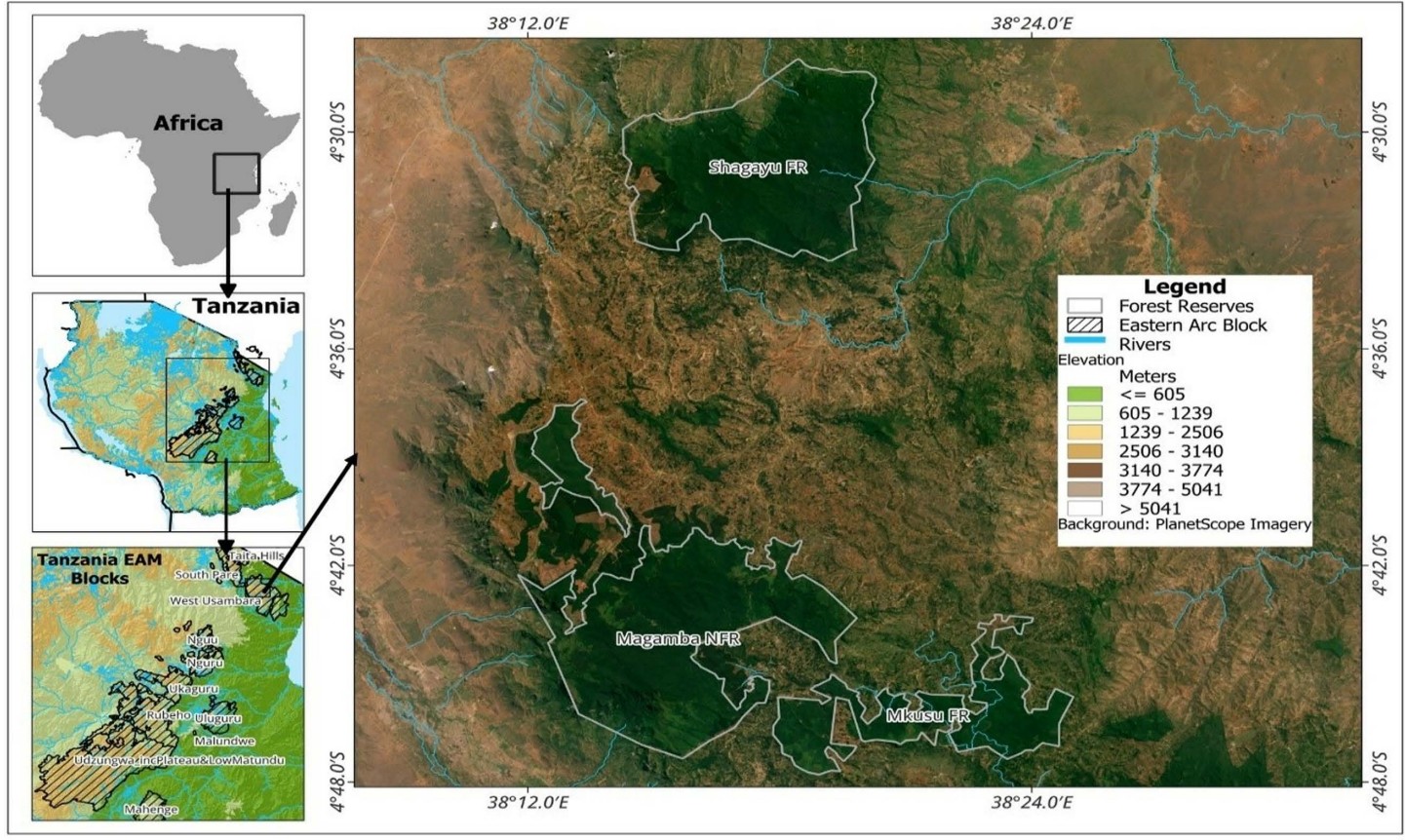

**Fig 1. The geographical location of Magamba NFR, Mkusu FR, and Shagayu FR in Lushoto District (Fig 1 was created using QGIS software.** The base map is from PlanetScope images via the Planet Explorer plugin (https://www.planet.com/explorer/) Digital Elevation Model(DEM) showing the elevation above sea levels in meters (m) is from USGS Earth Explorer (https://earthexplorer.usgs.gov/). Forest shapefiles are from UNEP-WCMC's Protected Planet (https://www.protectedplanet.net/en), and Eastern Arc Mountain Blocks shapefile is from the University of York (https://www.york.ac.uk/environment-geography/research/kite/resources/#tab-3).

Ferrasol for various activities and crops such as maize, beans, fruits, sweet and round potato, spices, and vegetables. Table 1 summarizes the geographical location of each study forest reserve.

## Research design

The study adopted a cross-sectional and mixed research design to collect qualitative and quantitative data through satellite images and socio-economic data, as presented in the methodological flowchart in Fig 2 below. Qualitative data was collected through PRA while quantitative data was collected using questionnaires. Key informant interviews, Focal Group Discussion (FGD), household interviews, and preference ranking (for matrix ranking) are among the PRA tools used.

**Table 1. Geographical location of MNFR, MFR, and SFR in Lushoto and Korogwe Districts.**

| Name of the FR | District | Size (Ha) | Latitude | Longitude | Altitude (m) | Rainfall (mm) | Temperature (⁰C) | GN |
|---|---|---|---|---|---|---|---|---|
| Magamba NFR | Lushoto & Korogwe | 9,283 | 38⁰15'E | 40⁰40'S | 2,800 | 1,200 | 22.5 | 103 |
| Mkusu FR | Lushoto | 3,674 | −4⁰46'E | 38⁰22'S | 1,600 | 900 | 26 | 114 |
| Shagayu FR | Lushoto | 7,830 | 38⁰16'E | 43⁰10'S | 1,668 | 1000 | 20 | SCH |

**Fig 2. Methodological flowchart depicting the link of geospatial and social-economic data to analyze forest fire burn severity in the West Usambara mountain forests.**

During the interview, 6 FGDs comprised 17 respondents from six villages to provide a comprehensive understanding of the issues related to forest fires and the management strategies.

Furthermore, this research adhered to ethical standards approved by the Directorate of Postgraduate Studies, Research, Technology Transfer, and Consultancy (DPRTC) of Sokoine University of Agriculture (SUA) under reference number DPRTC/R/186/35. Informed written and verbal consent was obtained from all participants, who were fully briefed on the study's purpose, focusing on forest fire sources, effects, and management, and assured of confidentiality and the right to withdraw at any time without penalty. All procedures were designed to minimize risks and protect participants' rights.

### Data collection methods

**Socio-economic data.** The field survey started on December 15, 2023, and ended on February 29, 2024, in Lushoto and Korogwe Districts. Participatory Rural Appraisal tools were used to gather social-economic data by developing semi-structured questionnaires [30]. Purposive sampling was used to select two villages from three pre-selected forest reserves based on the fire incidence and distance from the forest. Key informants including Magamba NFR staff, DFO, individuals knowledgeable about forest fires, environmental officers, village chairpersons, Village Executive Officers, and Village Natural Resource Committees (VNRC) were interviewed [30]. Additionally, questionnaires were administered to household heads during the data collection process covering general information on the local community perception on the sources and effects of fire, and how the fire regime has changed in the study areas in the last 10 years between 2013/2023. Respondents were determined based on the community who were directly involved in forest use, fire management, or handling fires. The sample size of 100 (10%) households was considered in the study, 62 were males and 38 were females from 1004 households in six study villages [31]. A checklist of questions was developed to guide the interview and discussion with the key informants. Also, household information regarding member's age, gender, marital status, income sources, educational level, land ownership, number of livestock, and employment were obtained.

Alternatively, the procedure for obtaining the household sample size in each village involved randomly selecting 10% of household heads from the total number of households (N) [32]. The total number of households (N) in each village was divided by 10% to determine the sample size for that specific village (Table 2).

Respondent's perception was elicited by obtaining their satisfaction ranking on the sources and effects of fire as an approach toward forest management. The respondents were specifically asked to rank the sources and effects of fire based on the Likert scale using five points ranging from 1 (Not known) to 5 (very well known) concerning the management aspects. The classification of topics based on the researcher's decision during the interview to focus on the key areas of importance related to forest fire management.

**Remote sensing data.** Multispectral data from two Landsat satellite system, namely Landsat 8 and 9 were used to map forest fires in the study area. Landsat-8 has a 16-day repeat period, a scan breadth of 185 km covering most of the

Table 2. The households and sample size.

| Forest Reserve | Ward name | Village name | Total population | No of Household | Village sample size 10% |
|---|---|---|---|---|---|
| Shagayu FR | Mlalo | Dule M | 952 | 81 | 8 |
| | Mtae | Sunga | 2,000 | 110 | 11 |
| Magamba NFR | Magamba | Magamba | 12,232 | 411 | 41 |
| | Mkumbara | Kwenangu | 3,657 | 131 | 13 |
| Mkusu FR | Migambo | Milungui | 6,447 | 210 | 21 |
| | Kwemashai | Kwemashai | 583 | 61 | 6 |
| **Overall Total** | | | **25,871** | **1,004** | **100** |

world, and a spatial resolution of 30 m [33]. It has four visible bands, one near-infrared (NIR) band, two shortwave infrared (SWIR) bands, two thermal bands (TIRS), one Cirrus band, and one panchromatic image [34,35]. The Landsat-9 OLI-2, the satellite in the Landsat series used in this study, was launched on September 27, 2021, and provides data continuity, it shares spectral ranges and geographic resolution with the previous Landsat-8 OLI sensor except for the thermal and panchromatic bands, which have a spatial resolution of 30 m. Landsat-9 OLI has 11 spectral bands [34]. Except for the thermal and panchromatic bands, which have a spatial resolution of 30 m, Landsat-9 OLI has 11 spectral bands. The spatial resolution of the panchromatic band is 15 m, while that of the thermal band is 100 m [36]. They are frequently employed in identifying vegetation change and forest fires [37].

The first step of the method consisted of gathering, extracting, and producing the necessary data sets of satellite images for the data research area obtained by GEE. In image selection, the seasonal occurrence of fire and the absence of clouds were taken into consideration during data collection on GEE for the study areas. For this purpose, image data-sets were collected before and after the fire for the period of 2013–2023. NBRpre, NBRpost, dNBR, NDVIpre, NDVIpost, and dNDVI indices were calculated on the images with the median statistics on the GEE cloud platform. Additionally, time series analysis was performed by calculating NDVI and NBR for each satellite image created.

The dNBR and dNDVI indices are highly effective for classifying forest fires due to their sensitivity in detecting burn severity, vegetation loss, and soil conditions before and after fires [38,39]. dNBR provides a continuous scale for assessing fire impacts, while dNDVI evaluates vegetation health, making them complementary tools for ecological monitoring [26]. These indices are reliable for identifying areas needing rehabilitation and informing fire management strategies. Their versatility is particularly valuable in mountainous areas like the West Usambara Mountain Forests, where traditional methods may be less effective.

Sentinel-2, with a 10-meter resolution and 5-day temporal frequency, enables timely detection and monitoring of fire events, capturing changes in burn severity and vegetation health shortly after fires [40]. Complementing this, Landsat 8 and 9, with a 30-meter resolution and 16-day intervals, are effective for post-fire analysis and long-term vegetation recovery monitoring (Table 3).

## Data analysis

**Socio-economic data.** Quantitative socio-economic data collected through questionnaires were coded and processed using IBM-SPSS version 27 statistical software, while the analysis was carried out through descriptive statistics and R software to generate frequencies and percentages presented in tabular forms. Qualitative data from the PRA exercise and key informants were analyzed by using content analysis to suffice qualitative evidence. Multiple response analysis was also performed to determine responses and percentages of respondents.

**Remote sensing data.** The satellite Data was analyzed using the GEE cloud platform. Since fire alters the spectral properties of the land surface by reducing vegetation and moisture content, therefore leads to decreased reflectivity in the visible and near-infrared wavelengths, while shortwave infrared reflectivity increases [25]. Several approaches used to assess forest burn mapping using satellite-based parameters were adopted to analyze forest fires [38]. The Normalised Burn Ratio

**Table 3. Date ranges for pre- and post-fire imagery used in composite imagery analysis.**

| Period | Pre-Fire Imagery Season | | Post-Fire Imagery Season | |
| | Start | End | Start | End |
|---|---|---|---|---|
| 2014 | 1 March | 30 July | 1 November | 30 February |
| 2017 | 1 March | 30 July | 1 November | 30 February |
| 2020 | 1 March | 30 July | 1 November | 30 February |
| 2023 | 1 March | 30 July | 1 November | 30 February |

(NBR) spectral index developed by the US Geological Survey (USGS) is a widely used approach for this purpose [24]. The NBR compares the near-infrared (NIR) and short-wave infrared (SWIR) reflectance values, with healthy vegetation showing high NIR reflectance and low SWIR reflectance [27]. This index is effective in distinguishing between areas affected by fire and areas with healthy vegetation based on their spectral characteristics and ranges from −1 to +1 [24].

$$NBR = \frac{NIR - SWIR}{NIR + SWIR}$$
(1)

In contrast to bare land and recently burned areas, high NBR values indicate healthy vegetation. Non-burned areas are normally attributed to values close to zero [38]. The difference between the pre-fire and post-fire NBR is used to calculate the delta NBR (dNBR), which can be used to estimate burn severity. The dNBR ranges from −2 to +2 with high positive values representing severely burned areas [24]. The higher dNBR values indicate more severe damage, while negative dNBR values indicate regrowth following a fire [27]. It is therefore the appropriate index for discriminating between burned and unburned areas, which contains information in the NIR and SWIR spectrum regions.

$$dNBR = NBR_{prefire} - NBR_{Postfire}$$
(2)

The dNBR index of the study area was calculated for two periods corresponding to the forest fires in 2013 and 2023. Therefore, four Landsat datasets (2014, 2017, 2020, and 2023) were used to get NBR and NDVI values (pre- and post-fires).

The NDVI can predict vegetation and biomass change pre-fire and post-fire, its values range between −1 and 1 [39]. Vegetated areas take positive values, while the negative values correspond to bare soils. High NDVI values represent dense green areas such as forests and cultivated areas [26]. NDVI equation was used to calculate the vegetation index to estimate the spatial and ecological biomass changes before and after the fire as well as an equation to obtain pre-fire and post-fire differences [39].

$$NDVI = \frac{NIR - RED}{NIR + RED}$$
(3)

$$dNDVI = NDVI_{prefire} - NDVI_{Postfire}$$
(4)

In this study, the dNDVI index was calculated by subtracting the pre- and post-fire NDVI values (Eq 4), and its values were reclassified into five classes (Fig 3).

Google Earth Engine (GEE) was used to process Landsat 8 and 9 imageries and examine the correlation between the Normalized Difference Vegetation Index (NDVI) and the Normalized Burn Ratio (NBR) [34]. The mean values of these indices were aggregated at 30-meter resolution and computed over the designated area of interest (AOI) [34,37,41]. For 2014, 2017, 2020, and 2023, we calculated the mean NDVI and NBR values throughout the AOI to create time series data. To make additional analysis easier, the datasets were combined into a single feature collection.

To measure the degree and direction of the linear relationship between NDVI and NBR, the Pearson correlation coefficient was computed [42]. The ee.Reduce.pearsonsCorrelation() function in GEE was used to derive the metric. To further understand the proportionate relationship between the indices, a linear regression model was fitted to the data using ee.Reducer.linearFit() to calculate the slope and intercept. The regression line and $R^2$ values for assessing the model fit were displayed. The results were then plotted in a scatter plot after the data shifted in R Studio.

The indices time series data for chosen years were extracted, and the region's mean values were combined in GEE. The mean values of NBR and NDVI were transformed into R studio and the ggplot2 package produced the line graphs used to illustrate trends and evaluate how vegetation as well as fire disturbance patterns changed over time.

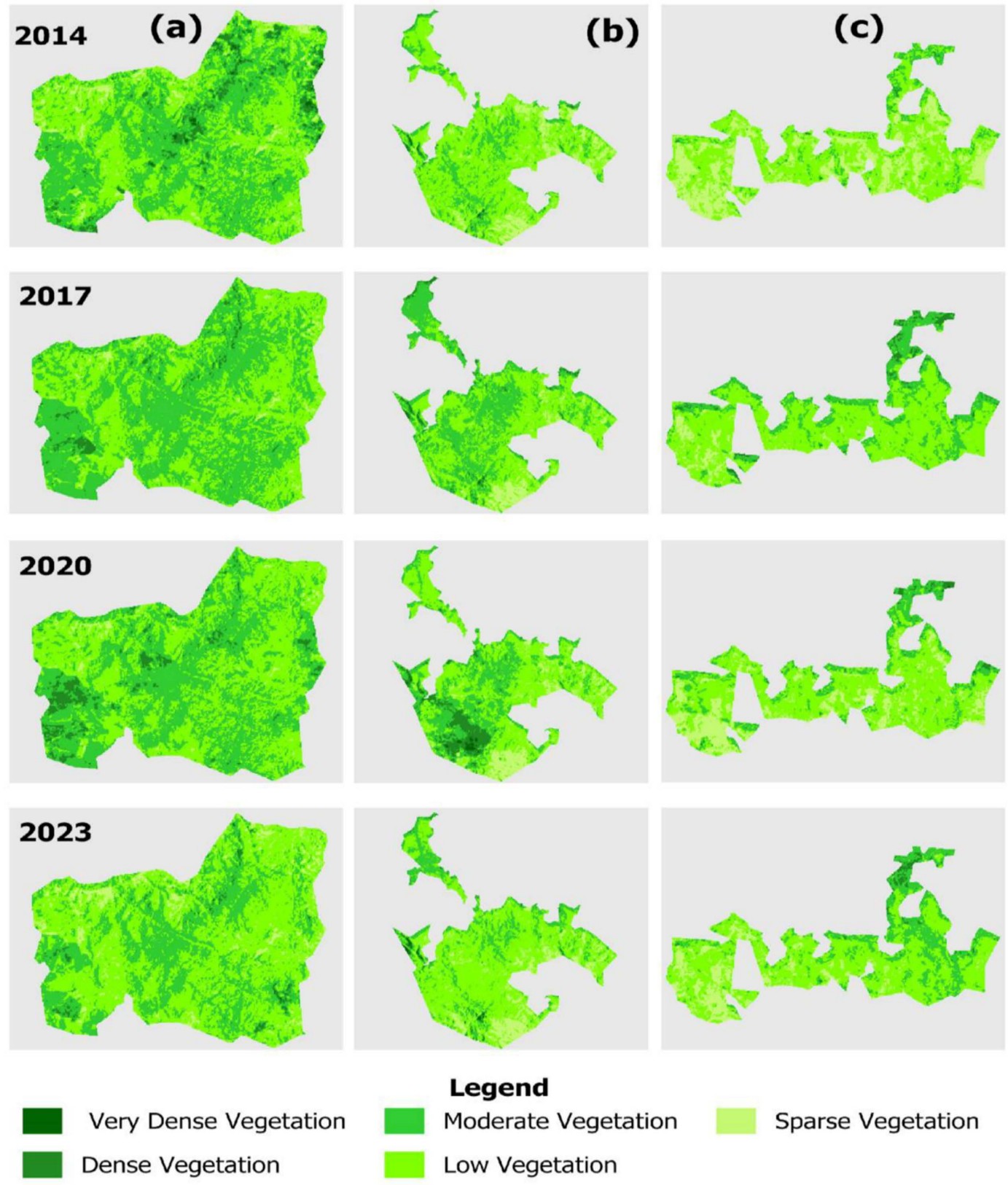

**Legend**

■ Very Dense Vegetation    ■ Moderate Vegetation    ■ Sparse Vegetation

■ Dense Vegetation    ■ Low Vegetation

**Fig 3. Spatial distribution of burn severity maps created according to NDVI for (a) Shagayu FR, (b) Magamba NFR, and (c) Mkusu FR (Figures are the output of Landsat 8 and 9 classified fire severity from the Google Earth engine that was exported into the Google Drive folder and imported in QGIS for visualization and spatial.** The grey color seen in the background is the background).

Visualization was carried out in QGIS software using data processed and mapped in google Earth Engine to ensure accurate spatial representation and analysis

## Results

### Socio-economic data

**Socio-economic characteristics of the respondents.** The average age of the respondents is between 31 and 40 years, with an annual household income of 2,422,216 TZA. About two-thirds (70.50%) of the participants were male and the remaining (29.50%) were female (Table 4). On the other hand, about 51.15% of the respondents completed primary school, 31.82% completed secondary school, 17.04% completed higher education, and none for informal education. Agriculture activities made up a vast majority (78.40%), followed by those working in beekeeping activities (8%), and the list one being mason (1.10%) as summarized in Table 4.

**General perception of local respondents on the source and effects of forest fires in the study area.** The sources of bushfires in Lushoto District are illustrated in Fig 4, with farm preparation identified as the leading cause at 38.2%, while local beliefs accounted for only 1.8%. Respondents ranked various effects of forest fires, with biodiversity loss being the most significant concern at 15.4%, followed by changes in forest status at 4% as summarized in Fig 5. Additionally, respondents ranked their perception and knowledge of forest fire from six topics describing how they understood and perceived the complexity of managing fire where (60%) of the respondents know the presented topics as illustrated in Fig 6.

### Remote sensing data

**Mapping burned areas.** Detecting forest fire areas using Landsat-8 and 9 satellite imagery and its vegetation indices has become a potential method for estimating forest fires [27]. The NBR index derived from the sentinel-2 satellite image was used to characterize burned areas and assess forest fire severity. At the same time, the NDVI variables confirm vegetation status within the forest over time. The NBR mean values ranged from 0.20 to 0.32 (Fig 7) and were categorized into five classes (low severity, high severity, unchanged, low regrowth, and high regrowth) (Fig 8). The spatial analysis from the map reveals that from 2013 and 2023 fire periods the study area was dominated by low (20.06%) and high (32.19%) burn severity. For this purpose, dNBR values were calculated from $NBR_{pre\text{-}fire}$ and $NBR_{post\text{-}fire}$ (Eq 2) using Landsat-8 and 9 images produced in the study area of 30 m spatial resolution from the GEE cloud platform.

NDVI maps are classified as very dense vegetation, dense vegetation, moderate vegetation, low vegetation, and sparse vegetation (Fig 3). Additionally, the non-burned areas from the 2013 and 2023 fires represented 11.47%, 20.4%, and 30.09% of the study area in Magamba NFR, Mkusu FR, and Shagayu FR respectively (Fig 9).

The total burned area was 3,296.96 ha, constituting 15.86% for Magamba NFR, Mkusu FR, and Shagayu FR (Fig 9). Areas with low and high regrowth and unchanged account for 84.14% of all total areas. In Magamba NFR, the fire affected 1322.80 ha, which is 14.25% of the total area, with low and high-severity fires impacting 1272.05 ha and 50.75 ha respectively. Mkusu FR experienced a total burned area of 1341.23 ha, which is 36.50% of the total area with high and low-severity fires affecting 108.88 ha and 1232.35 ha respectively. Shagayu FR had a burned area of 632.93 ha, which is 8.1% of the total area with 575.04 ha and 57.89 ha for low and high-severity fires respectively. Mkusu FR was identified as the most affected reserve, while Shagayu FR was the least affected.

The correlation between the NBR and NDVI was computed in the Google Earth Engine in each Forest Reserve. Their values for each point were then computed using R-studio ggPlot2 package to plot the scattered charts shown in Figs 10–12.

**Table 4. Socio-demographic characteristics.**

| Characteristic | Category | Frequency | Percentage | Mean |
|---|---|---|---|---|
| Sex | Male | 62 | 70.5 | |
| | Female | 26 | 29.5 | |
| Education level | Primary education | 45 | 51.2 | |
| | Secondary level | 28 | 31.8 | |
| | Higher education | 15 | 17 | |
| | Informal education | 0 | 0 | |
| Occupation | Agriculture | 69 | 78.4 | |
| | Beekeeping | 7 | 8 | |
| | Livestock keeping | 4 | 4.5 | |
| | Petty business | 6 | 6.8 | |
| | Carpenter | 1 | 1.1 | |
| | Mason | 1 | 1.1 | |
| Age | | | | 40.84 |
| Household income | | | | 2422215.9 |

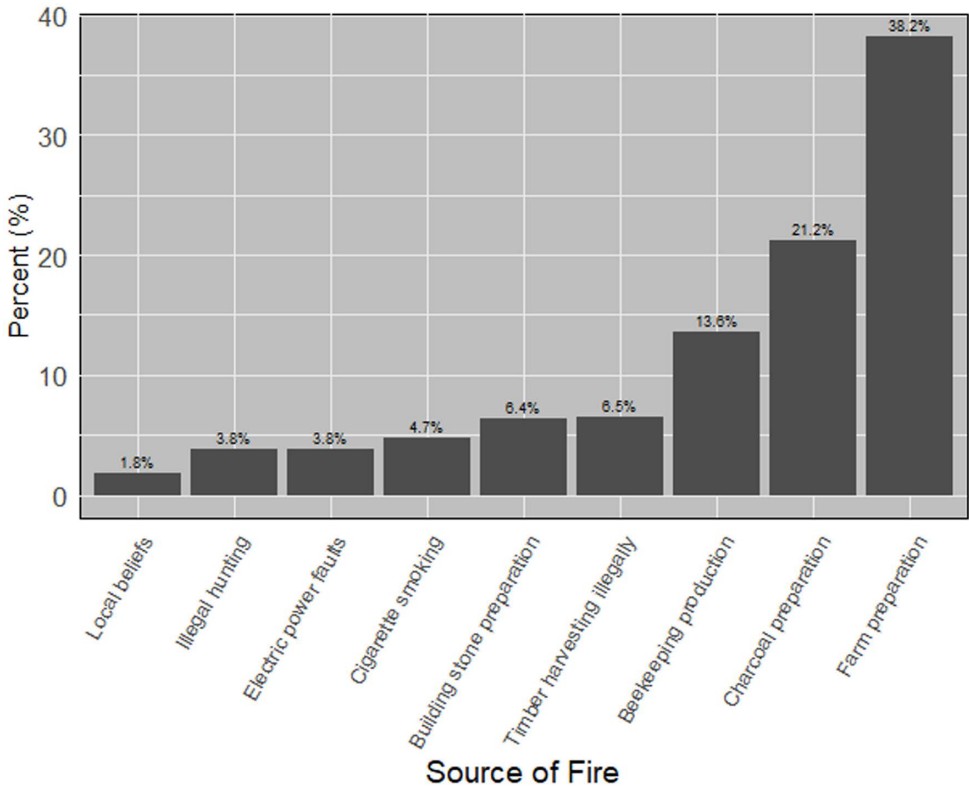

**Fig 4. Perception of local respondents on the source of forest in the study area.**

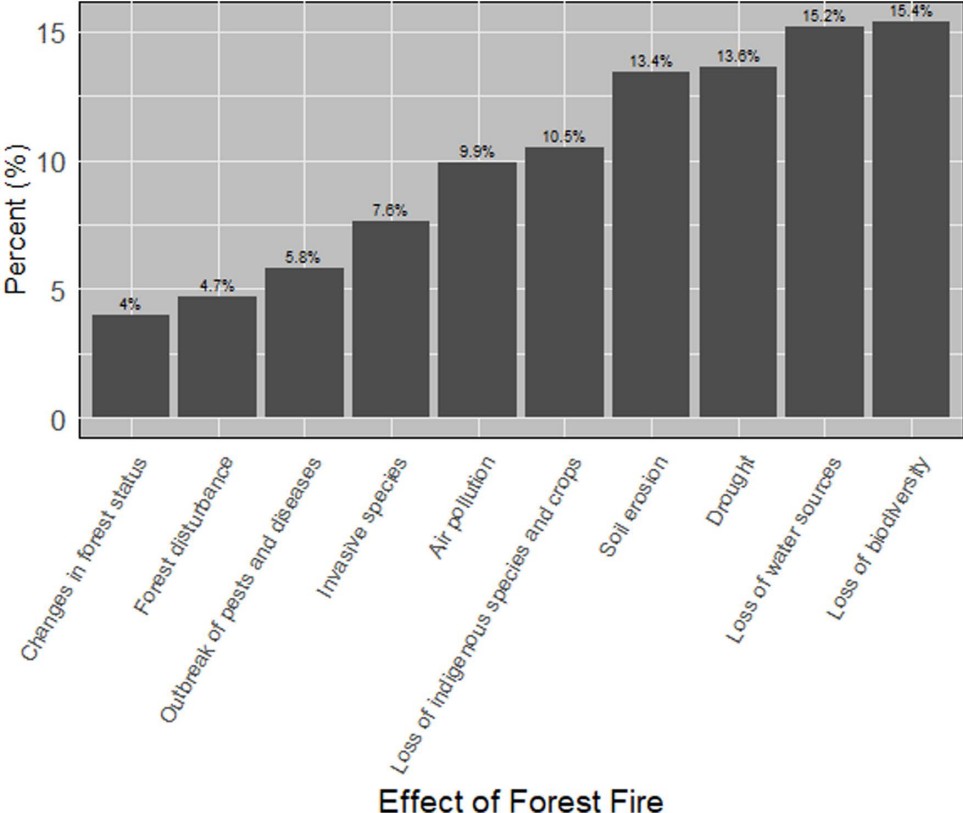

**Fig 5. Perception of local respondents on the effects of forest in the study area.**

Table 5 shows the correlation coefficients R (0.92, 0.96. and 0.98) and their respective $R^2$ values for Magamba NFR, Mkusu FR, and Shagayu FR respectively. These statistics values indicate strong positive linear correlations between NBR and NDVI indices, which are compared to assess fire severity and vegetation condition derived from RS data and field observations. Shagayu FR shows the strongest relationship, followed by Mkusu FR, and then Magamba NFR.

Meanwhile, when assessing the fire severity and vegetation health between NBR and NDVI indices in Figs 10–12 it is evident that as NBR values increase, NDVI values also tend to rise in both study areas. It suggests that as areas recover from fire damage (indicated by higher NBR values), the health of vegetation (as reflected by NDVI) also improves. Therefore, the strong relations between NBR and NDVI suggest using these spectral indices for burn severity estimation and monitoring post-fire recovery. The mean NBR values indicate the severity of the burn while the mean NDVI values provide insight into the health and recovery of vegetation. For example, in Fig 7 the NBR mean values of Magamba NFR, Mkusu FR, and Shagayu FR were 0.26, 0.24, and 0.25 respectively.

Likewise, NDVI and NBR mean values in Magamba NFR, Mkusu FR, and Shagayu FR were 0.31, 0.34, and 0.30 respectively (Fig 7). The high NDVI mean values indicate dense, healthy vegetation while the low NDVI mean value reflects sparse or unhealthy vegetation.

**NBR and NDVI time series analysis.** The time series assessment of both NBR and NDVI performed in the GEE platform often shows similar trends with little deviation with some points in response to fire events. For instance, both indices typically exhibit a decline immediately after the fire (2015, 2017, 2018, and 2024 for both MFR and MNFR), followed by gradual recovery as vegetation begins to regrow between 2016, 2019–2023 (Figs 13–15).

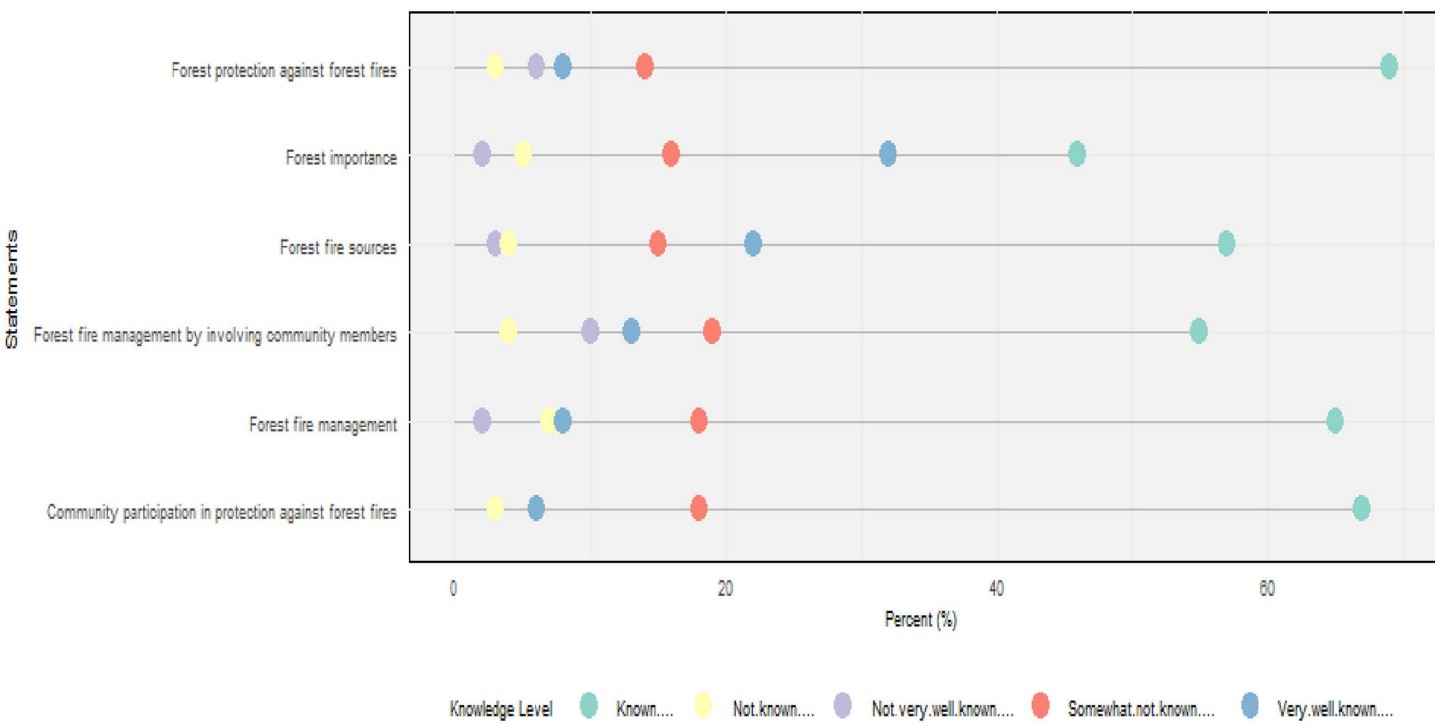

**Fig 6. Likert scale rating regarding perception and knowledge topics on forest fire from respondents.**

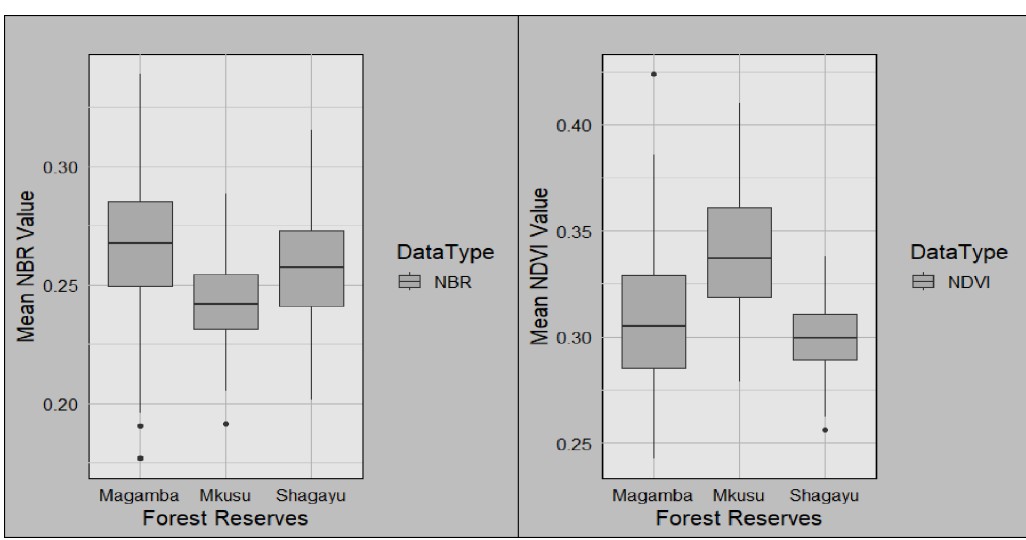

**Fig 7. The NBR and NDVI mean values for Magamba NFR, Mkusu FR, and Shagayu FR.**

## Discussion

The study aimed to develop an integrated approach using geospatial tools and socio-economic data to assess the sources and effects of forest fires and map their burn severity in the West Usambara Mountain Forests. RS and

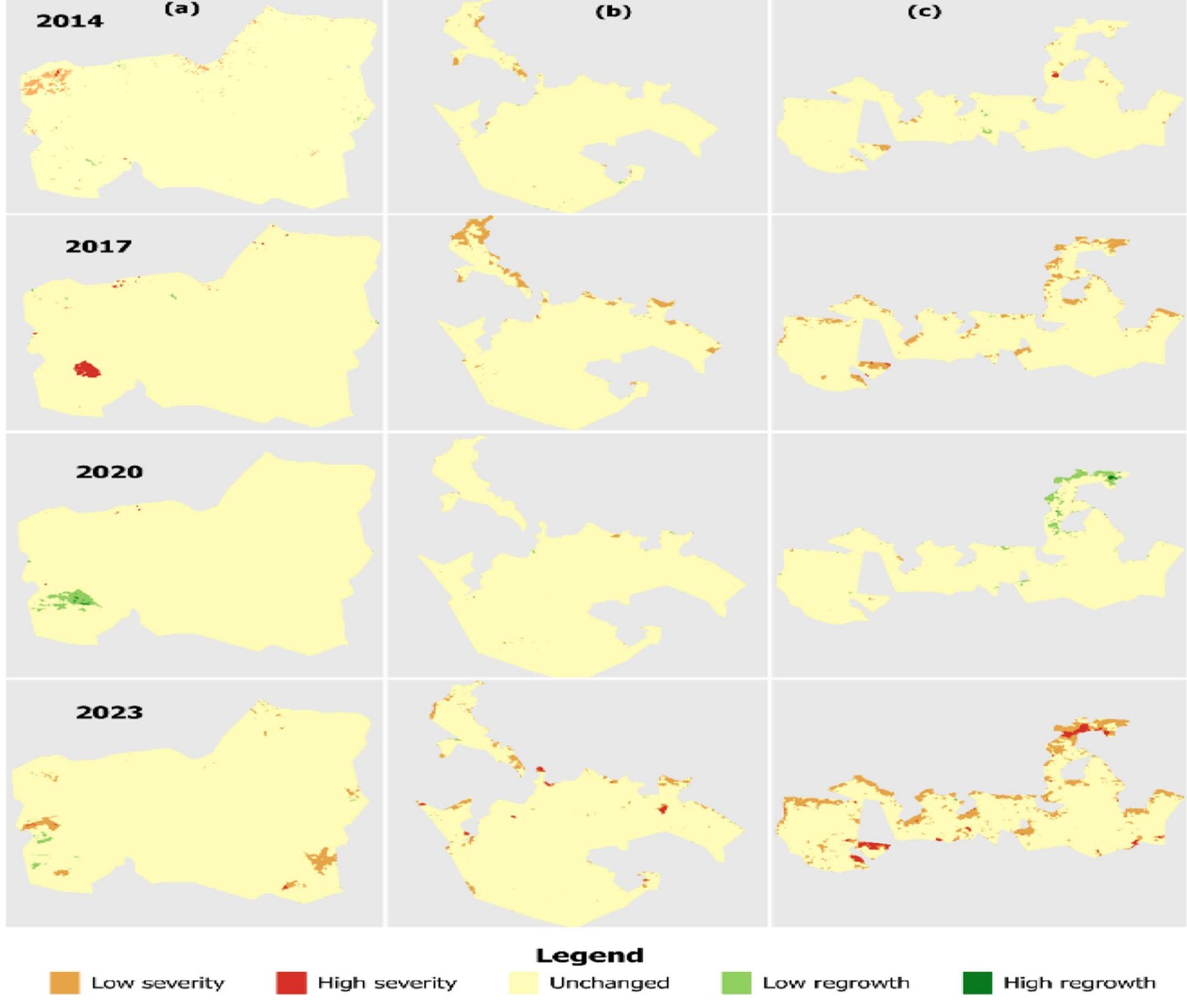

**Fig 8. Spatial distribution of burn severity maps created according to NBR for (a) Shagayu FR, (b) Magamba NFR, and (c) Mkusu FR (Figure are the output of Landsat 8 and 9 classified fire severity from the Google Earth engine that was exported into the Google Drive folder and imported in QGIS for visualization and spatial.** The grey color seen in the background is the background).

socio-economic data reveal that agriculture and charcoal production are the primary sources of forest fire burn severity, aligning with the findings of [43].

The findings of this study indicate that the average age of the respondents between 31 and 40, reveals a significant presence of young, financially capable individuals. At the same time, those with 60 and above are less active economically [30]. Meanwhile, African traditional gender roles highlight distinct social roles assigned based on gender, where men typically head households [44]. Additionally, higher education levels are associated with increased access to technical

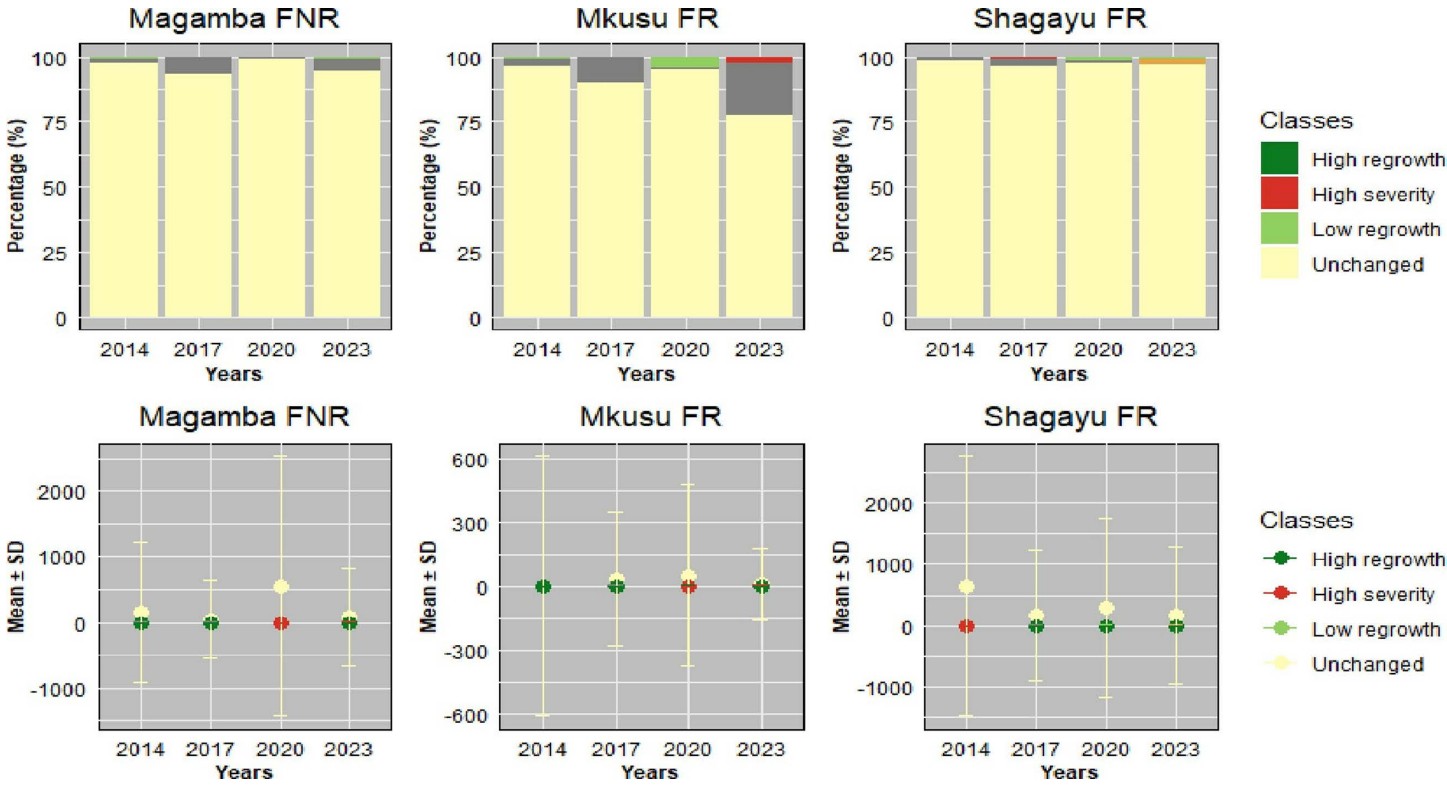

**Fig 9. Area changes over time for Magamba NFR, Mkusu FR, and Shagayu FR.**

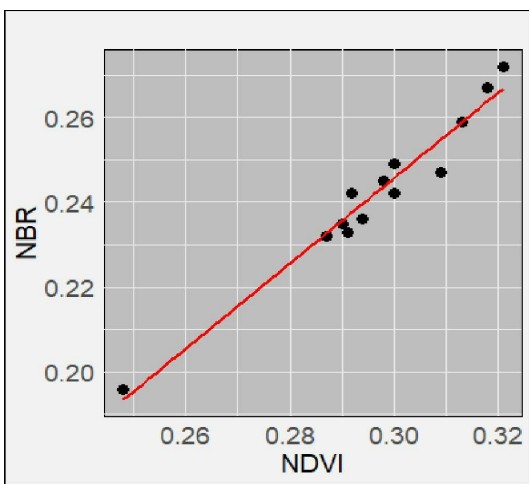

**Fig 10. Correlation analysis between NBR and NDVI for Shagayu FR.**

information on fire management, enabling individuals to adopt innovative and sustainable practices [45], as it is hypothesized that as education levels rise, more respondents will embrace sustainable fire use practices advocated by CBFiM [46]. Other studies have also found that older people perceive conservation positively as a gift for future generations [47].

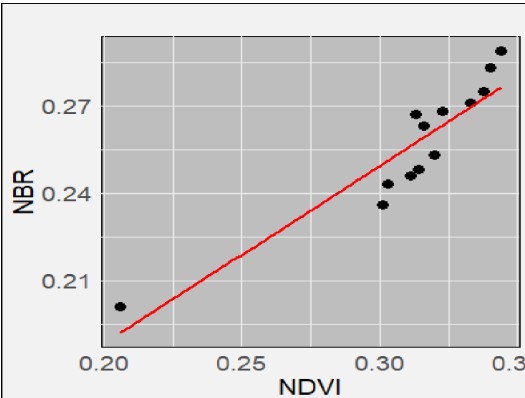

**Fig 11. Correlation analysis between NBR and NDVI for Magamba NFR.**

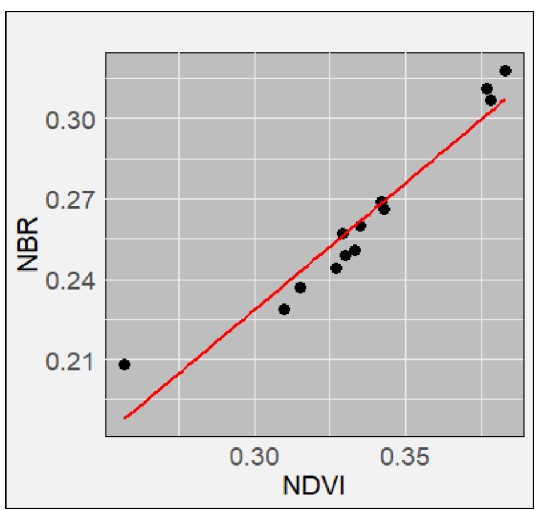

**Fig 12. Correlation analysis between NBR and NDVI for Mkusu FR.**

**Table 5. Correlation coefficient (R) and coefficient of determination (R²) values in the study area.**

| Forest name | Correlation Coefficient (R) | Coefficient of Determination (R²) |
|---|---|---|
| Shagayu | 0.98 | 0.96 |
| Magamba | 0.92 | 0.84 |
| Mkusu | 0.96 | 0.92 |

The widespread practice of zero grazing in the study area suggests community awareness of land management practices [30]. However, using fire for farming and charcoal production contributes significantly to forest fires [11,18]. Forest fires adversely impact biodiversity, which is vital to the community, affecting traditional knowledge, economic stability, food security, and cultural practices, as reported by key informants. The perception of more high-intensity fires in the study area over the past decade is supported by the literature on the case of increased burned areas in northwest Portugal [48]. These fires also influence ecosystem services, recreational values, and educational opportunities [18]. In particular,

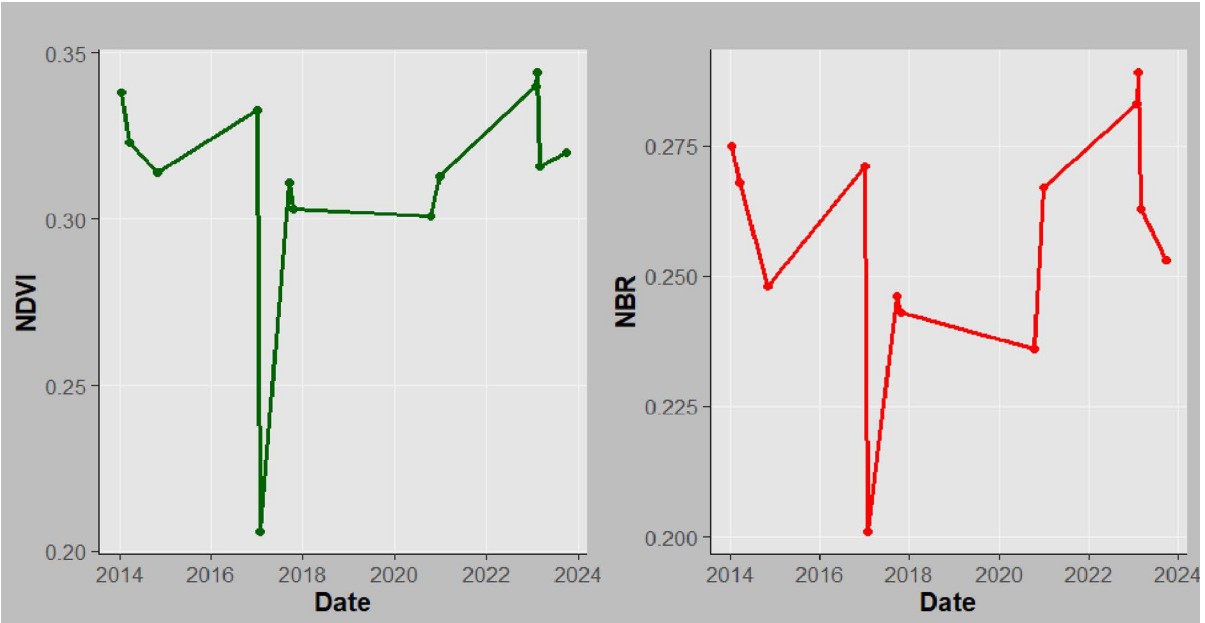

**Fig 13.  NBR and NDVI time series for Mkusu FR.**

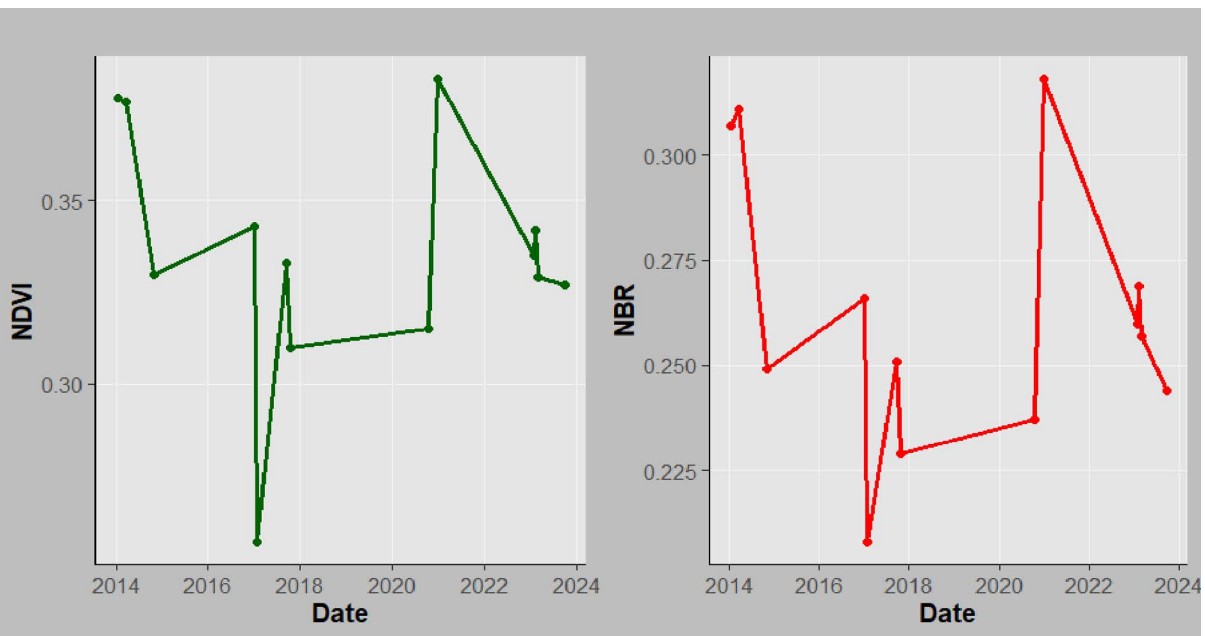

**Fig 14.  NBR and NDVI time series for Shagayu FR.**

agriculture is a significant source of income for villagers in the study area, with common crops including maize, beans, potatoes, and vegetables. The prevalent practice of slash-and-burn agriculture, especially in areas experiencing population growth, significantly contributes to the occurrence of forest fires in tropical fires [14,18,49].

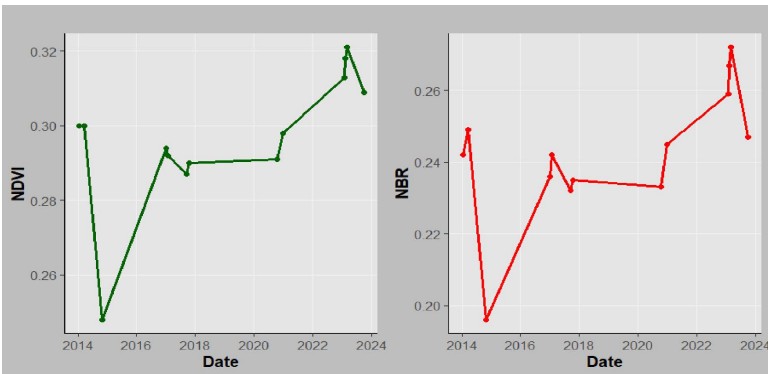

**Fig 15. NBR and NDVI time series for Magamba NFR.**

Furthermore, charcoal production and cultural beliefs about fire use can contribute to forest fires, although these are less prevalent in the study area. For example, in Sunga village, one informant stated that *"Sambaa tribe believes that if one starts a fire and it ends up getting and spreading to a large extent, signals a long life."* Local respondents recognize the importance of fire for various activities but also acknowledge the devastating consequences of unmanaged fires, such as loss of biodiversity and forest degradation [18]. Despite community awareness and education efforts, high-fire incidents continue to occur in national forest reserves in Tanzania [50]. According to the key informants, the availability of significant tangible benefits such as economic incentives from eco-tourism, water for irrigation, and domestic use are the reasons for active willingness to be involved in forest fire protection similar to those of these scholars [51,52]. Lack of or insufficient tangible benefits from national forests can lead to disappointment among local people [53].

Findings from the Remote Sensing component of this study confirm that reliable burn severity estimates can be derived from combined indices of NBR and NDVI for mapping burn severity and vegetation status. For example, [54] found that both Sentinel-2 and Landsat-8 provided similar estimates of burn severity, which aligns with the findings of this study. Through NDVI and NBR indices, we tested and compared for burn severity mapping to justify the results (Figs 3 and 8), as [55] did in their study research. The NBR index records the effect of fire with enhanced spectral contrast, making it easier to estimate burned areas, assess burn severity, and monitor vegetation recovery after a fire [24]. This study demonstrated high accuracy in detecting fire-affected areas using the NBR and NDVI indices derived from Sentinel-2 images, as supported by multiple research findings [24,38,39]. Similarly, the NDVI was used to detect the extent of vegetation status and destruction of existing vegetation before and after the fire as supported by [56]. When comparing NBR and NDVI results, the NDVI results analysis observed that areas burned transformed into open spaces or bare areas within the forest after the fire as supported by [56]. According to [57], the healthier and lusher the vegetation, the higher the corresponding NDVI value. [26] used the dNBR index derived from Sentinel-2 images to detect forest fires similar to the approach taken in this study. While dNBR is effective for detecting burned areas, it is sensitive to water, which can lead to misclassification of high-severity pixels; however, water masking was applied in the study area to address this issue [58]. Additionally, [59] mapped burned areas using the Burning Area Index (BAI) and NBR to detect fire-affected regions in Mediterranean countries while for this study, these indices demonstrated excellent results, proving to be effective in tropical and mountainous areas as well.

Consequently, despite the numerous studies that show strong correlations between field-based reference observation of burn severity and both post-fire and differenced spectral indices [39,60,61], the sensitivity of these methods varies across different vegetation types [56], so they need to be field-validated to ensure that reliable information is generated. The strong correlation enhances the reliability of using these indices for effective fire management and assessment with

$R^2$ values reflecting the effectiveness of RS data in accurately estimating fire effects [27]. The strong correlation between NBR and NDVI values typically indicates a strong positive correlation as vegetation health improves (higher NDVI), and the area is likely to recover from fire damage (higher NBR) [62]. On the other hand, the mean NBR values indicate the severity of the burn while the mean NDVI values provide insight into the health and recovery of vegetation [27]. The high NBR mean values indicate healthy or unburned vegetation while the low NBR mean value indicates that the area has experienced significant fire damage [39]. Also, the high NDVI mean values indicate dense, healthy vegetation while the low NDVI mean value reflects sparse or unhealthy vegetation.

Similarly, findings further show that a sharp decrease in NBR and NDVI in 2017 for MFR and SFR indicates forest burning during that period. This correlation reinforces the reliability of using both indices for monitoring fire effects and assessing vegetation status [39]. The high correlations suggest that changes in one index can help to infer changes in the other index, making them valuable tools for forest fire management and ecological studies [58]. Additionally, NDVI time series analysis is used to track changes in vegetation, particularly in areas that have been rehabilitated following wildfires [39]. It is commonly used to identify vegetation density changes before and after fire. The observed decrease in NDVI in 2017 in this study aligns with [26] who found the same scenario in Arouca. [63] reported that post-fire NDVI values decreased to 0.36 in areas with high burn severity while for this case, a sharp decrease up to 0.21 in Mkusu FR was observed. [64] examined the dynamic behavior of pre and post-fire vegetation in a burned forest area through NDVI time series analysis and they found a sharp decrease in post-fire NDVI values in Northern Italia. Also, [65] found similar results for Madeira Island when detecting forest fire severity and vegetation health over time using NBR and NDVI indices also similar to this study. In this study, the areas where the fire occurred are highly dominated by Blacken ferns (*Pteridium aquilinum*) which immediately become dominant in burn areas after fire [19]. After the fire, NDVI values were between 0.21–0.36, indicating a decrease in the dense vegetation. Therefore, low NBR and NDVI mean values show substantial burn and loss of vegetation health and density in some periods as the study aligns with [39].

## Conclusion

The present study integrates geospatial tools and socio-economic data to identify forest fire burn severity zones, sources of fire, and their effects using Sentinel-2 satellite data and the Google Earth Engine (GEE) cloud platform to create thematic layers of burn severity and ultimately produce a comprehensive synthetic burn severity maps. The use of the GEE platform in this study significantly enhances geospatial data processing capabilities, advancing research on forest fire prediction. GEE provides a fast, reliable, and effective means of producing burn severity maps, offering substantial advantages for monitoring pre and post-fire conditions, particularly in large and mountainous forest areas. Remote Sensing findings indicate that NBR and NDVI indices effectively estimate burn severity and give clear vegetation status. On the other hand, socio-economic data indicates that incorporating stakeholder perceptions into management decisions enhances societal acceptability and effectiveness, with the community identifying shifting cultivation and charcoal production as the primary sources of forest fires. Therefore, the study provides a basis for future fire management and prevention efforts, emphasizing the need for awareness regarding the sources and effects on livelihood and the effective control mechanisms. They also highlight the critical importance of burn severity mapping for informed decision-making and strategic planning in both high and low-risk zones of the west Usambara Forest in Lushoto, Tanzania.

The study recommends enhancing community participation in forest conservation by increasing local knowledge and awareness of Community-Based Fire Management (CBFiM) and Integrated Fire Management (IFM) approaches. This involvement fosters a sense of ownership and shared responsibility, addressing encroachment and managerial challenges. But also the use of advanced remote sensing technologies is highly encouraged to accurately assess burn severity and monitor changes over time. Future research should focus on exploring studying the long-term impacts of forest fires on biodiversity and ecosystems, including the effect on

soil health and vegetation recovery patterns in the Western Usambara Mountain forests and inform management practices.

## Acknowledgments

The authors would like to acknowledge all participants who contributed their insights, particularly Mr. Hassan Sengerere, a botanist and expert in Inventory exercise from Tanzania Forest Services Agency (TFS) in Lushoto. We also extend our gratitude to the Tanzania Forest Services Agency (TFS) Headquarters and President's Office, Regional Administrative and Local Government (PO-RALG), for providing necessary data collection permits.

## Author contributions

**Conceptualization:** Braison Paul Paul Mkiwa, Ernest W. Mauya, Gimbage E. Mbeyale.

**Data curation:** Braison Paul Paul Mkiwa.

**Formal analysis:** Braison Paul Paul Mkiwa, Justo N. Jonas.

**Funding acquisition:** Braison Paul Paul Mkiwa, Ernest W. Mauya.

**Investigation:** Braison Paul Paul Mkiwa.

**Methodology:** Braison Paul Paul Mkiwa, Ernest W. Mauya, Gimbage E. Mbeyale.

**Resources:** Braison Paul Paul Mkiwa, Ernest W. Mauya.

**Software:** Braison Paul Paul Mkiwa, Ernest W. Mauya, Justo N. Jonas.

**Supervision:** Ernest W. Mauya, Gimbage E. Mbeyale.

**Visualization:** Braison Paul Paul Mkiwa.

**Writing – original draft:** Braison Paul Paul Mkiwa.

**Writing – review & editing:** Braison Paul Paul Mkiwa, Ernest W. Mauya, Gimbage E. Mbeyale.

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
