## [Decision Letter · Decision Letter 0]

11 Oct 2024

PONE-D-24-41614Integrating geospatial tools in mapping forest fire severity and burned areas in the Western Usambara Mountain Forests, Lushoto, TanzaniaPLOS ONE

Dear Dr. Mkiwa,

Thank you for submitting your manuscript to PLOS ONE. After careful consideration, we feel that it has merit but does not fully meet PLOS ONE’s publication criteria as it currently stands. Therefore, we invite you to submit a revised version of the manuscript that addresses the points raised during the review process.

The reviews, included at the bottom of the letter, indicate that your manuscript could be suitable for publication following revision. We hope that you will consider these suggestions, and revise your manuscript.Recommendation: Major Revision

We look forward to receiving your revised manuscript.

Kind regards,

Mitiku Badasa Moisa

Academic Editor

PLOS ONE

Journal Requirements:

2. You indicated that ethical approval was not necessary for your study. We understand that the framework for ethical oversight requirements for studies of this type may differ depending on the setting and we would appreciate some further clarification regarding your research. Could you please provide further details on why your study is exempt from the need for approval and confirmation from your institutional review board or research ethics committee (e.g., in the form of a letter or email correspondence) that ethics review was not necessary for this study? Please include a copy of the correspondence as an ""Other"" file.

" I have read the journal's policy and the authors of this manuscript have the following competing interests: Reports of financial support provided by the Tanzania Forest Services Agency (TFS) and Tanzania Forest Fund (TaFF)."

"The authors thank the Tanzania Forest Services Agency (TFS) and Tanzania Forest Fund (TFF) for financial support for field work and manuscript development."

"B.P.Mkiwa - Name of the author received the award

Grant number: Not specified.

Names: Tanzania Forest Services Agency (TFS) and

             Tanzania Forest Fund (TaFF)

TFS URL: http://www.tfs.go.tz.

TFF URL:https://www.mfukowamisitu.go.tz

NO-The sponsors or funders did not play any role in the study design, data collection and analysis, decision to publish, or preparation of the manuscript"

6. We note that your Data Availability Statement is currently as follows: All relevant data are within the manuscript and its Supporting Information files.

8. We note that Figures 1 and 7 in your submission contain map/satellite images which may be copyrighted. All PLOS content is published under the Creative Commons Attribution License (CC BY 4.0), which means that the manuscript, images, and Supporting Information files will be freely available online, and any third party is permitted to access, download, copy, distribute, and use these materials in any way, even commercially, with proper attribution. For these reasons, we cannot publish previously copyrighted maps or satellite images created using proprietary data, such as Google software (Google Maps, Street View, and Earth). For more information, see our copyright guidelines: http://journals.plos.org/plosone/s/licenses-and-copyright.

a. You may seek permission from the original copyright holder of Figures 1 and 7 to publish the content specifically under the CC BY 4.0 license.  

Additional Editor Comments:

The reviews, included at the bottom of the letter, indicate that your manuscript could be suitable for publication following revision. We hope that you will consider these suggestions, and revise your manuscript.

Recommendation: Major Revision

Reviewers' comments:

Reviewer's Responses to Questions

**Comments to the Author**

1. Is the manuscript technically sound, and do the data support the conclusions?

Reviewer #1: No

Reviewer #2: Yes

Reviewer #3: Yes

2. Has the statistical analysis been performed appropriately and rigorously? 

Reviewer #1: No

Reviewer #2: Yes

Reviewer #3: Yes

3. Have the authors made all data underlying the findings in their manuscript fully available?

Reviewer #1: Yes

Reviewer #2: Yes

Reviewer #3: Yes

4. Is the manuscript presented in an intelligible fashion and written in standard English?

Reviewer #1: No

Reviewer #2: Yes

Reviewer #3: Yes

5. Review Comments to the Author

Reviewer #1: Critical Comments for Improvement

Abstract

Clarity and Conciseness: The abstract should clearly summarize the key objectives, methods, findings, and implications in a more concise manner. Currently, it may feel too detailed for an abstract.

Quantitative Highlights: Include specific quantitative results, such as the percentage of areas affected by fires, in the abstract to provide immediate insight into the study's significance.

Implications: Briefly mention the practical implications of the findings in the abstract to highlight their relevance to stakeholders.

Introduction

Literature Review: The introduction should include a more comprehensive literature review on forest fires in Tanzania and the use of geospatial tools. This contextualizes the study within existing research and highlights gaps that the study addresses.

Research Questions: Clearly state the specific research questions or hypotheses at the end of the introduction to guide the reader on what to expect.

Local Context: Provide more background on the socio-economic context of the Western Usambara Mountain area, including the local communities' reliance on forest resources, which will strengthen the rationale for the study.

Methodology

Method Justification: Explain why specific geospatial tools and methodologies (e.g., dNBR, dNDVI) were chosen over other potential methods. This can enhance the credibility of the approach.

Sampling Details: Clarify the sampling strategy for the Participatory Rural Appraisal (PRA) and how participants were selected. This will help assess the representativeness of the data.

Temporal Analysis: Provide more detail on the temporal resolution of satellite imagery used and how it aligns with fire events. This is crucial for understanding changes over time.

Limitations: Discuss any potential limitations in the methodologies used, such as uncertainties in remote sensing data or challenges in ground validation.

Results

Data Presentation: Use clearer visuals, such as graphs or maps, to present key findings. This can enhance comprehension and highlight important trends more effectively.

Statistical Analysis: Provide more detailed statistical analyses to support claims about correlations or trends observed in the data.

Detailed Findings: Expand on the findings related to the severity of fires, perhaps including more categories of severity or the ecological impact of different levels of burn severity.

Discussion

Linking Results to Literature: Better integrate the results with existing literature, discussing how the findings align or contrast with other studies on forest fires in similar contexts.

Broader Implications: Discuss the broader implications of the findings for forest management and policy in Tanzania, including specific recommendations based on the results.

Future Research Directions: Suggest areas for future research that could build on the study’s findings, such as long-term ecological impacts of fires or the effectiveness of different fire management strategies.

Socio-economic Considerations: More deeply explore the socio-economic implications of the findings, particularly regarding local communities' dependence on forest resources and potential conflicts arising from fire management practices.

Integration of Local Knowledge: Emphasize the importance of integrating local knowledge into fire management strategies, which could lead to more effective and culturally relevant solutions.

By addressing these critical comments, the study can significantly enhance its clarity, depth, and relevance, making it a valuable contribution to the field of forest fire management in Tanzania

Reviewer #2: Here are some major comments and suggestions for each section of the manuscript on integrating geospatial tools for mapping forest fire severity and burned areas in the Western Usambara Mountain Forests:

Abstract

Clarity and Conciseness: Ensure the abstract succinctly summarizes the study's objectives, methods, key findings, and significance. Avoid jargon and keep it accessible.

Results Highlight: Include specific quantitative results to give readers a clear idea of the study's impact.

Key Terms: Consider defining critical terms (e.g., "burn severity") for clarity.

Introduction

Context and Rationale: Provide more background on the significance of forest fires in the Usambara Mountains. Why is this region particularly vulnerable?

Literature Review: Incorporate recent studies on forest fire mapping and severity assessment to highlight gaps your research addresses.

Research Objectives: Clearly articulate the research questions or hypotheses guiding the study. Ensure these are explicitly linked to the issues raised in the introduction.

Methodology

Geospatial Tools: Detail the specific geospatial tools and software used (e.g., remote sensing, GIS applications) and justify their selection.

Data Sources: Clarify the sources of data (e.g., satellite imagery, ground truth data) and their relevance to your study.

Analysis Techniques: Provide more detail on the analytical methods (e.g., algorithms for assessing burn severity) and how they were implemented.

Limitations: Acknowledge any limitations of your methodology that could affect results (e.g., resolution of data, seasonal variations).

Results

Presentation of Data: Use clear visuals (maps, graphs) to present findings effectively. Ensure figures are well-labeled and referenced in the text.

Interpretation of Results: Offer insights into what the data reveals about fire severity and burned areas. Discuss any unexpected findings.

Statistical Analysis: If applicable, include details on statistical tests performed and their significance.

Conclusions

Summary of Findings: Recap the main findings succinctly, emphasizing their implications for forest management and policy.

Future Research: Suggest areas for future research based on your findings, particularly in relation to fire prevention and ecosystem recovery.

Practical Applications: Discuss how this research could inform local conservation efforts or influence fire management strategies in the region.

Overall Comments

Cohesiveness: Ensure each section flows logically into the next, maintaining a consistent narrative throughout the manuscript.

Language and Style: Check for grammatical accuracy and stylistic consistency. Aim for a formal yet engaging tone.

References: Ensure all cited works are up-to-date and relevant, following the appropriate formatting style.

Reviewer #3: Research on assessing the Integrating geospatial tools in mapping forest fire severity and burned areas was a current issue and highly innovative finding for the forest resource management activities. However, I have some concerns about the current research and will give some suggestions on how the manuscript may be improved for publication as follows:

1. On the tittle

On the cover page, the tittle of the manuscript deals about integrating geospatial tools in mapping forest fire severity and burned areas in the Western Usambara Mountain Forests. But, in the body of the manuscript, the locations of the study area were classified into different study village which may confuse the readers. So, tittle modifications should required.

2. In the abstract

Please elaborate the targeted model you used for data analysis for your study.

On the line 15 and 16, the findings revealed that agricultural activities (44.5%) and charcoal production (21.1%) are the primary causes of these fires doesn’t explain the reality exist on fig.3a. you have to refer and edit it accordingly.

3. In the introduction It is better if you add

the scientific definitions of forest fire

Lists of some native forests to the study area and a detail history about the forest fire before your study period in the study area.

Short and understandable research gap for your study.

Generalized forms of your study objective rather than the specific one.

4. Materials and methods

Re-edit the geographical locations of the study area according the figure 1 reading.

Why don’t you specify your total study area? it was the mandatory to now the effects of forest fire on the total study area over the study period.

Please, specify the forest covers of the study area as the main description of the study area.

5. Data sources

It was better if you put your geospatial data in tabular form with detail descriptions of data source, types, special resolution, specific purposes and downloaded year and time.

Specify the formula you used to limit your sample size for respondents nomination from household for interview.

How many Focus Group Discussions (FGDs) you were organized during your study to summarize the whole idea raised from the respondents?

6. Results

According to the results displayed on figure 4, the respondent’s perception and knowledge about forest fire were classified into six categories. So, what was your base for this classification?

It was better if you increase and improve the visibility of the results presented on the figure 3a, b and 4 for the readers.

7. Discussion and recommendation

Your discussion was organized and addressed the whole contents of your finding in comparison with pre conducted researches. However, your recommendation mainly for local community only. Are sure that local community alone can manage the effects of forest fire without involvements of governmental institutions?

6. PLOS authors have the option to publish the peer review history of their article (what does this mean? ). If published, this will include your full peer review and any attached files.

**Do you want your identity to be public for this peer review?** For information about this choice, including consent withdrawal, please see our Privacy Policy .

Reviewer #1: No

Reviewer #2: No

Reviewer #3: No

---

## [Author Response · Author response to Decision Letter 1]

23 Jan 2025

Response to the editor and reviewers are included in the attachment.

---

## [Decision Letter · Decision Letter 1]

28 Apr 2025

Integrating geospatial tools in mapping forest fire severity and burned areas in the Western Usambara Mountain Forests, Lushoto, Tanzania

PONE-D-24-41614R1

Dear Dr. Mkiwa,

We’re pleased to inform you that your manuscript has been judged scientifically suitable for publication and will be formally accepted for publication once it meets all outstanding technical requirements.

Kind regards,

Bijay Halder

Academic Editor

PLOS ONE

Additional Editor Comments (optional):

Reviewers' comments:

Reviewer's Responses to Questions

**Comments to the Author**

1. If the authors have adequately addressed your comments raised in a previous round of review and you feel that this manuscript is now acceptable for publication, you may indicate that here to bypass the “Comments to the Author” section, enter your conflict of interest statement in the “Confidential to Editor” section, and submit your "Accept" recommendation.

Reviewer #1: All comments have been addressed

Reviewer #3: All comments have been addressed

2. Is the manuscript technically sound, and do the data support the conclusions?

Reviewer #1: Yes

Reviewer #3: Yes

3. Has the statistical analysis been performed appropriately and rigorously? 

Reviewer #1: Yes

Reviewer #3: Yes

4. Have the authors made all data underlying the findings in their manuscript fully available?

Reviewer #1: Yes

Reviewer #3: Yes

5. Is the manuscript presented in an intelligible fashion and written in standard English?

Reviewer #1: Yes

Reviewer #3: Yes

6. Review Comments to the Author

Reviewer #1: Dear Authors,

Congratulations on your valuable contribution to the field. I am pleased to inform you that all the previous comments and suggestions have been thoroughly addressed in the revised manuscript. Your responses were clear and constructive, and the revisions have significantly improved the quality and clarity of the work.

I have no concerns regarding dual publication, research ethics, or publication ethics in your manuscript.

I recommend the manuscript for publication in its current form.

Reviewer #3: All comments have been addressed and have not further comments. I accepted for publication in this journal

7. PLOS authors have the option to publish the peer review history of their article (what does this mean? ). If published, this will include your full peer review and any attached files.

**Do you want your identity to be public for this peer review?** For information about this choice, including consent withdrawal, please see our Privacy Policy .

Reviewer #1: **Yes: ** Zenebe Reta Roba

Reviewer #3: No

---

## [Editor Report · Acceptance letter]

PONE-D-24-41614R1

PLOS ONE

Dear Dr. Mkiwa,

I'm pleased to inform you that your manuscript has been deemed suitable for publication in PLOS ONE. Congratulations! Your manuscript is now being handed over to our production team.

Kind regards,

on behalf of

Mr. Bijay Halder

Academic Editor

PLOS ONE